# Criteria for Cutting Head Clogging Occurrence during Slurry Shield Tunneling

**Xinggao Li** [1,2] **, Yi Yang** [1,2,*] **, Xingchun Li** [3] **and Hongzhi Liu** [4]

1 Key Laboratory of Urban Underground Engineering of the Education Ministry, Beijing Jiaotong University, Beijing 100044, China; lixg@bjtu.edu.cn
2 School of Civil Engineering, Beijing Jiaotong University, Beijing 100044, China
3 Faculty of Intelligent Manufacturing, Wuyi University, Jiangmen 529020, China; 13116360@bjtu.edu.cn
4 CCCC Tunnel Engineering Company Limited, Beijing 100102, China; liuhongzhi@ccccltd.cn
* Correspondence: 17121139@bjtu.edu.cn

**Abstract:** Cutting head clogging is more frequently encountered as more tunnels are being excavated by slurry shield machines. So establishing criteria for cutting head clogging occurrence based on the machine driving parameters is of great engineering significance. Three construction cases of the Beijing south-to-north water diversion auxiliary project, the Jinan Huanghe River Crossing tunnel construction, and the Wuhan Metro Line 8 Yangtze River Crossing tunnel construction, are introduced. Development of the main driving parameters in the construction cases, including the total thrust, the cutting head torque, the advance rate, and the cutting head rotation speed of the tunneling machines before, during, and after the cutting head clogging, are presented and analyzed. The fact is that the total thrust and the cutting head torque of tunneling machines will increase, or will not, once the cutting head clogging occurs. It is recommended to take two combined parameters of total thrust/penetration depth and cutting head torque/penetration depth into account to judge whether the cutting head clogging will occur or not. The maximum increases of the composite parameters by 2–6 times are found in the construction cases. But for the minimum increase, a 30–50% increase of the composite parameter should be noted. The findings can be of great help for similar projects.

**Keywords:** slurry shield tunneling; cutting head clogging; driving parameters; criterion





## 1. Introduction

In recent years, closed-face tunneling machines, namely earth pressure balance shield machines and slurry shield machines, are being widely used to excavate tunnels in soft ground because of their many merits, such as safety, high advance rate, and reduced disturbance to the surrounding ground. When tunneling in fine-grained soils, excavated materials sometimes adhere to the cutting head and/or the conveyor system of the tunneling machines, which is referred to as clogging [1,2] or adhesion [3,4]. These issues, if not properly dealt with, will cause excavation difficulties, the extension of the construction duration, and consequentially result in budget overrun.

The problem of clogging in soft ground shield tunneling has aroused widespread concern, and numerous efforts have been devoted to it. Because clogging is concerned with both cohesion of soil particles and the adhesion of the soil to a metal surface, existing studies have attempted to evaluate it in two ways. The first approach of estimating the clogging potential is dependent on the indices of plasticity and consistency of the involved soils, where the plasticity index is defined as the difference in moisture content of soils between the liquid and plastic limits expressed in percentage, and the consistency index $I_c$ is defined as:

$$I_c = (w_L - w)/I_p \tag{1}$$

where the parameter $w$ represents actual water content, $w_L$ is the liquid limit, and $I_p$ is the plasticity index. This approach is particularly beneficial for practical purposes,

and many researches focused on it. Thewes and Burger (2004) [1] presented a clogging diagram only by judging the properties of an in situ clay regarding the particular operating conditions of a slurry-supported shield. Based on a combined evaluation of water content and plasticity, Hollmann and Thewes (2013) [5] developed a new diagram to evaluate the clogging potential of clay soils for all types of tunneling machines, and this new diagram enabled the assessment of the consequences when the water content of a clayey soil is changed during excavation. The second approach is based on assessing soil adhesion to the metal surface using direct shear tests, vane shear tests, and other laboratory tests. For example, Zimnik et al. (2000) [6] investigated the adherence behavior of clay by measuring the shear stress required to shear the clay over a steel surface in a direct shear box, and results showed a great influence of the steel roughness, contact time and mineral type. Using kaolin clay, Sass and Burbaum (2009) [7] did tests to simulate clay sticking to the cutting wheel of the machine to evaluate the likelihood of adhesion potential, and the results of tests suggested that the adhesion changed with the compression force. Other factors, including clay mineralogy and roughness of the face of the cutting wheel, were also considered in the tests. Feinendegen et al. (2011) [8] developed a new classification scheme for clogging potential based on the cone pull-out test to detect the adhesion/clogging propensity of a soil or rock. Zumsteg and Puzrin (2012) [9] proposed new methods for stickiness and adhesion quantification, and a newly developed device was used to measure the tangential adhesion and sliding resistance between a steel plate and soft soil pastes at different applied pressures. Basmenj et al. (2016) [10] performed normal piston pull out and modified direct shear tests to determine the normal or tangential adhesion stress of the clayey soil samples. The results indicated that normal adhesion values showed meaningful variation with consistency index and were compatible with the base of field clogging assessment criteria. Based on discussing the laboratory tests for assessing clogging potential, Thewes and Hollmann (2016) [11] explained a diagram for assessing clogging risks for all types of tunneling machines, and introduced a new testing scheme for the evaluation of sedimentary rocks regarding clogging. Zumsteg et al. (2016) [12] investigated the effects of slurry on the stickiness of excavated clays of the slurry shield tunneling, and found the stickiness of the excavated soil correlated with the shear resistance of the slurry. As a result, increased slurry strength, while beneficial for excavations in coarse soils, may increase susceptibility to clogging under mixed face conditions. Basmenj et al. (2017) [13] investigated the adhesion potential of kaolinite and montmorillonite using pull-out tests, and the results revealed differences between minerals concerning adhesion value and behavior because of their differences in the microstructural properties and dominant sticking mechanism of each clay mineral. Burbaum and Sass (2017) [3] conducted experiments to analyze the influence of two elements, i.e., the pore water tension (suction) of the soil and the permeability of the soil, on the adhesion of cohesive soils to solid surfaces, and presented the results of adhesion tests and pore water tension measurements during the adhesion tests. Alberto-Hernandez et al. (2018) [2] provided a state of the art report on the clogging potential of tunneling machines, and existing research work on mechanisms of clogging, factors affecting the soil adhesion to a steel surface, laboratory tests to assess clogging potential, clogging potential diagrams, and mitigation and countermeasures were well summarized. Alberto-Hernandez et al. (2018) [14] indicated that the initial water content, the roughness of shear plate, and the percentage of additives have a significant effect on clogging potential, and an empirical diagram to understand the clogging potential by relating the clogging potential to soil properties was also plotted.

The occurrence of the cutting head clogging is closely related to the physical and mechanical properties of the soils in the cutting face. However, the soils encountered in each cycling of the soil cutting and segment assembly of the close face shield tunneling cannot often be determined beforehand due to the irregular changes of the natural ground. The methodology for evaluating clogging potential relying only on the soil types is sometimes unrealistic and impractical. The clogging effect continues to be a serious issue in tunnels construction in clayey soils because the current criteria for evaluating clogging oc-

currence have proven to be only partially effective due to the unknown soils and unknown properties of the soils. Notwithstanding, once cutting head clogging happens, the shield machine performance parameters such as cutter head torque, total thrust, advance rate, etc., will experience rapid changes. Usually, the cutting head torque and the total thrust of the machine increase, and the advance rate and the penetration depth per revolution decrease. Therefore, establishing criteria for cutting head clogging during soft ground shield tunneling based on the machine's main driving parameters is feasible and of great engineering significance. This approach for evaluating cutting head clogging based on the key shield machine performance parameters is validated in three shield tunneling projects. Figure 1 shows the geological map where the three shield tunnel cases are located.

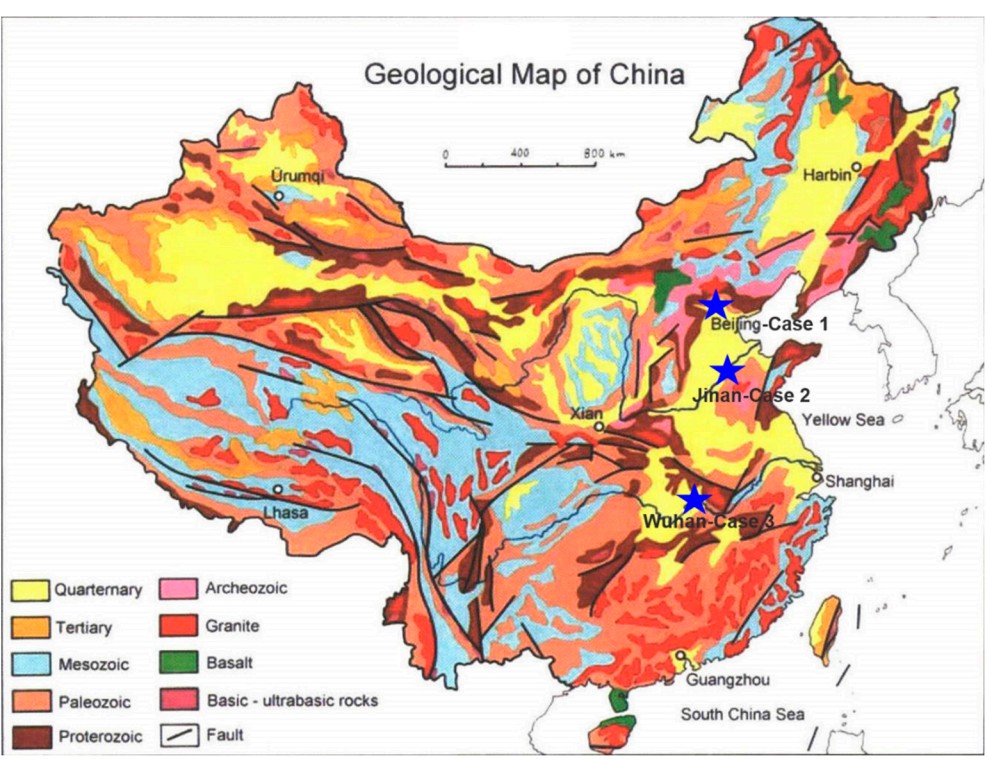

**Figure 1.** Geological overview map where the location of the three cases.

## 2. Cutter Head Clogging at the Beijing South-to-North WATER Diversion Auxiliary Project

The second phase project of the Tuan-Jiu Part of the Beijing south-to-north water diversion auxiliary project includes a 1.7 km long tunnel. The tunnel excavated by a slurry shield is 6 m in outer diameter and 5.4 m in inner diameter. As shown in Figure 2, the involved soils at the job site between the mileage of 2 + 202 and the mileage of 3 + 050 mainly consist of cobble ④, clay ⑤, and cobble ⑥. Both the cobble ④ layer and the cobble ⑥ layer are well graded with medium sand filling. The cobble ④ layer whose cobble content is 60–70% is around 15–18 m in thickness, with an uttermost grain size of 2–5 cm and a maximum grain size of 16 cm. The cobble ⑥ layer with a cobble content of 55–65% is about 8–10 m in thickness with a grain size of 4–6 cm. The clay ⑤ layer is mostly 1–2 m thick, and the maximum thickness is 4 m. The clay ⑤ layer with a natural moisture content of 31.8%, plasticity index of 19.2, cohesion of 48.8 kPa, and internal frictional angle of 12.1° is mostly 1–2 m in thickness, and the maximum thickness is 4 m.

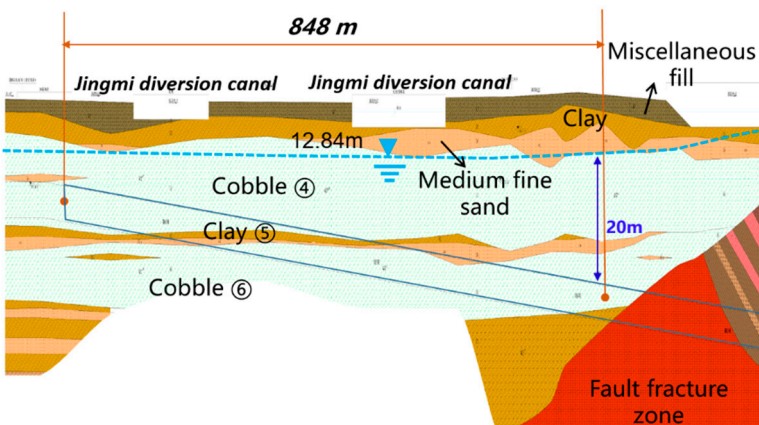

**Figure 2.** Involved soils of the tunnel construction at the Beijing south-to-north water diversion auxiliary project.

The groundwater is about 12.8 m underneath the ground surface. The used slurry shield is equipped with a plate-type cutting head, as presented in Figure 3. The opening area ratio of the cutting head is 32% on the whole, and a slightly higher ratio of 35% at the cutting head center area. 

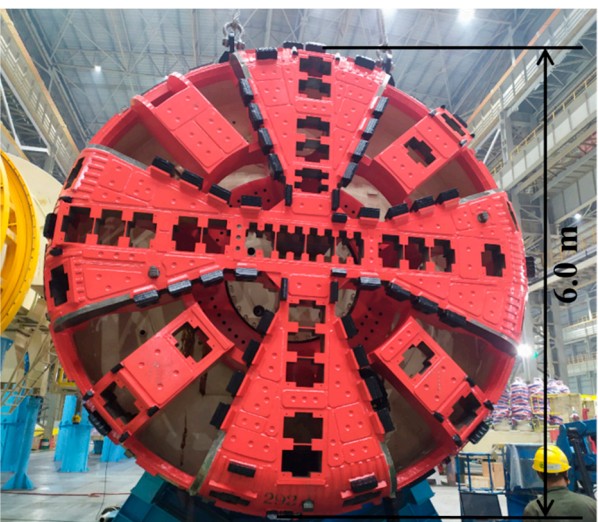

**Figure 3.** The used cutting head at the Beijing south-to-north water diversion auxiliary project.

When shield tunneling in cobble-clay mixed face ground condition, no countermeasures were taken beforehand due to the insufficient information regarding the distribution of clay soils. The cutting head clogging initiated from ring 145, resulting in the rapid changes of the driving parameters, as shown in Figures 4 and 5. The clay-cobble mixture presented in Figure 6 and collected at ring 145 form the discharged muck showed cutting head clogging occurrence.

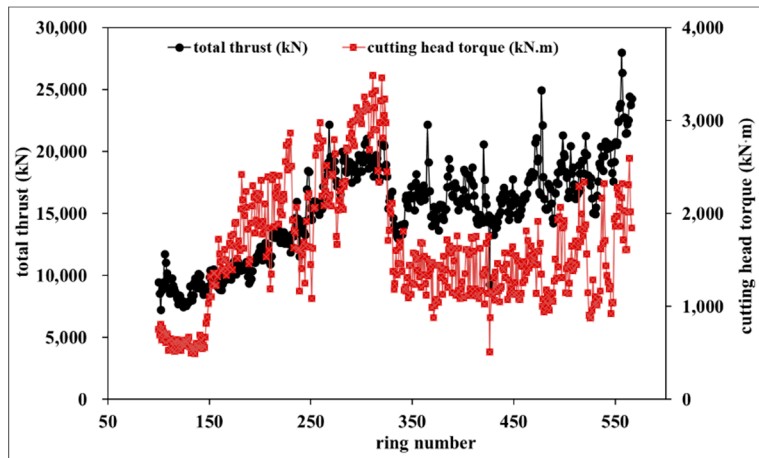

**Figure 4.** Changes of total thrust and cutting head torque of the machine at the Beijing south-to-north water diversion auxiliary project.

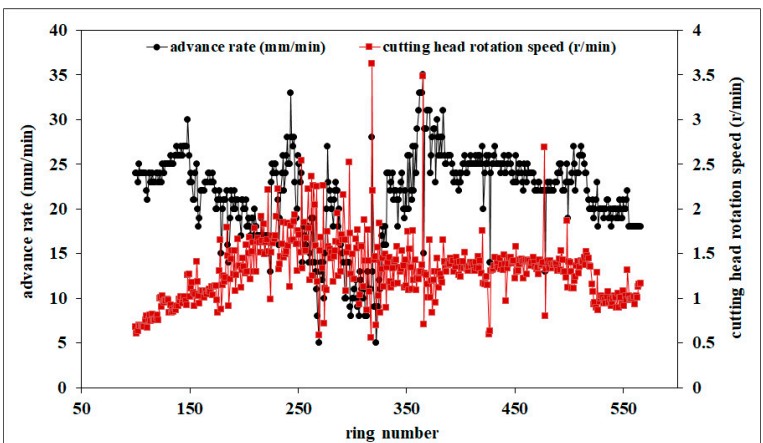

**Figure 5.** Changes of advance rate and cutting head rotation speed of the machine at the Beijing south-to-north water diversion auxiliary project.

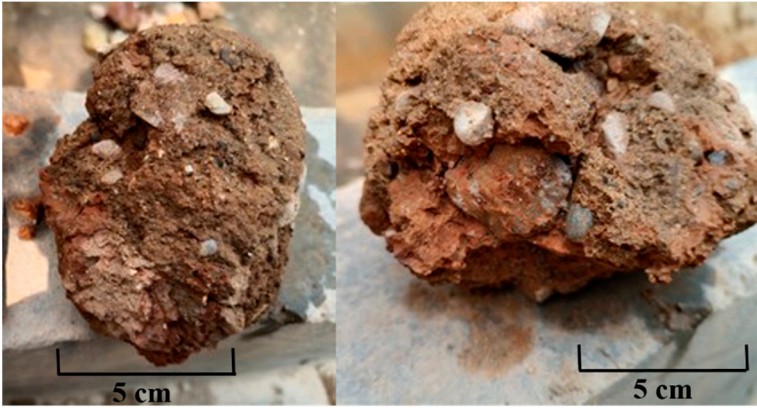

**Figure 6.** Clay-cobble mixture collected at the ring 145 at the Beijing south-to-north water diversion auxiliary project.

Before ring 145, the machine driving parameters remained basically stable with a total thrust of 7200–12,000 kN, a cutting head torque of 500–900 kN·m, a machine advance rate of almost 25 mm/min, and the cutting head rotation speed of 0.7–1.0 r/min. Due to the cutting head clogging, the driving parameters experienced remarkable changes after ring 145. Sharp increases of the total thrust and the cutting head torque were found with a total

thrust of more than 20,000 kN and the cutting head torque over 3500 kN·m at the ring 320. At rings 311 and 322, an anti-clogging agent was added to the excavation chamber to soften the mud cake in and on the cutting head. Between rings 320 and 340, the total thrust and the cutting head torque decreased gradually to 13,000 kN and 1300 kN·m, respectively. After ring 340, an increase was not seen with the total thrust and the cutting head torque fluctuating around 17,000 kN and 1500 kN·m, respectively. Between ring 145 and ring 320, the advance rate of the machine decreased from 25 mm/min to 5 min/min due to cutting head clogging, and the cutting head rotation speed increased gradually to 2 r/min. In the following rings, the advance rate rose to over 30 mm/min, firstly because of the adding of the anti-clogging agent into the excavation chamber, and then declined to 25 mm/min little by little or even less at the end. The cutting head rotation speed was increased to more than 2 r/min, then gradually decreased to 1.5 r/min or even less.

For a tunneling shield machine, the cutting head torque is closely related to the total thrust, the advance rate, and the cutting head rotation speed, and is a combined result of many influencing factors [15]. Cutting head torque increases due to the cutting head clogging, but the torque increase does not mean the occurrence of clogging. For example, the increase of the penetration depth per revolution of the cutting head, which is defined as the ratio of advance rate to cutting head rotation speed, can also lead to the increase of the cutting head torque. This is true of the total thrust of the machine. So two composite parameters are introduced: total thrust/penetration depth and cutting head torque/penetration depth, which are plotted in Figure 7. From the development of the two new parameters, the clogging at ring 145 and the effects due to the countermeasures taken at rings 311 and 322 can be more clearly and directly presented. The newly-defined parameters magnify the side effects of the cutting head clogging on shield machine driving, which can be used to evaluate the occurrence of the cutting head clogging.

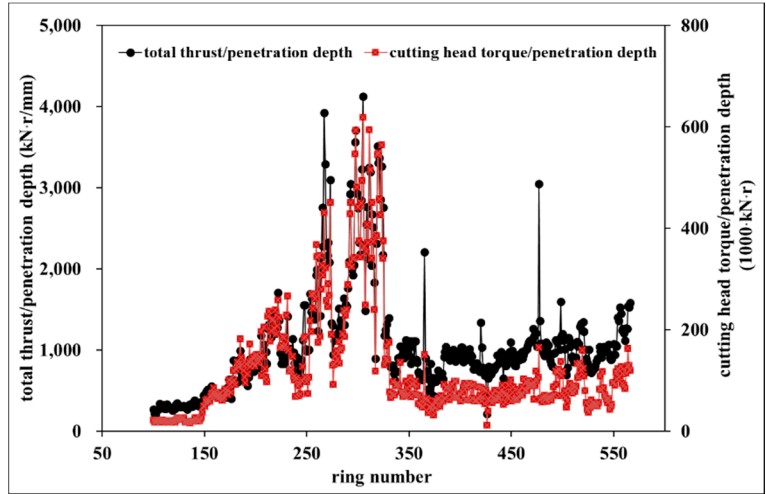

**Figure 7.** Changes of the two composite parameters at the Beijing south-to-north water diversion auxiliary project.

## 3. Cutter Head Clogging at the Jinan Huanghe River Crossing Tunnel Construction

The two Jinan Huanghe River Crossing tunnels, located at the central area of the Jinan city of Shangdong Province of China, were excavated by two super-diameter slurry shields. The shield machines were 15.76 m in excavation diameter with an opening area ratio of 46%, as shown in Figure 8.

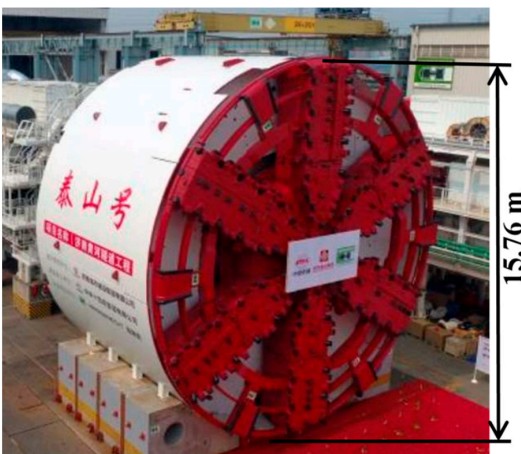

**Figure 8.** The used shield machine at the Jinan Huanghe River Crossing tunnel.

The two tunnels at a depth of 26.3–54.6 m, and the soils to be excavated are mainly silty clay with a clay grain content of 15.5–45.5% (average of 26.9%). Sand layers and calcic concretions are within the tunnel alignment but unevenly distributed due to local enrichment and stratification, as shown in Figure 9. The encountered silty clay with a plasticity index of 10–15 and a consistency index of 0.5–1.0 is liable to clogging the cutting head. The Huanghe River is at the ground surface, and the possible water pressure during the construction is about 0.65 MPa. As presented in Figures 10 and 11, when shield driving at ring 675 of the east line tunnel, the total thrust and the cutting head of the machine experienced a sharp increase, and the advance rate began to decrease. The increase and the decrease in the main driving parameters continued until ring 685.

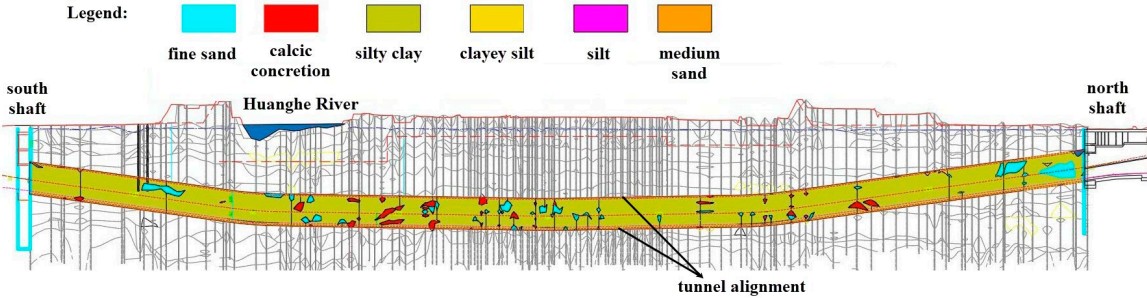

**Figure 9.** Longitudinal cross section of the Jinan Huanghe River Crossing tunnel.

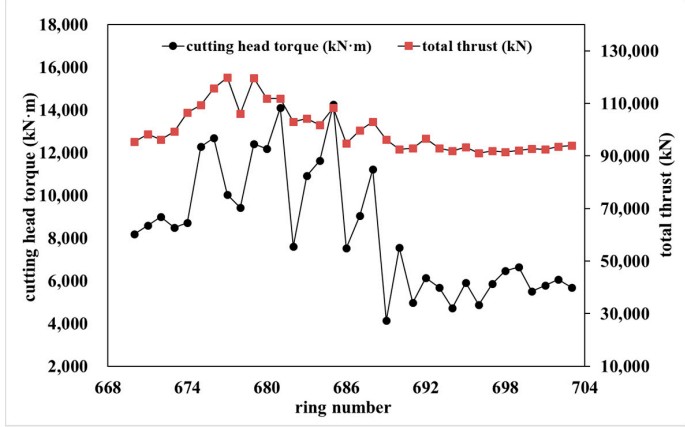

**Figure 10.** Changes of total thrust and cutting head torque of the machine at the Jinan Huanghe River Crossing tunnel construction.

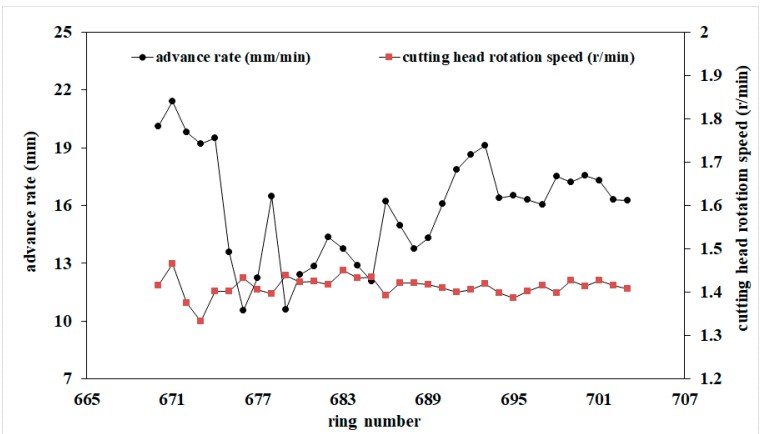

**Figure 11.** Changes of advance rate and cutting head rotation speed of the machine at the Jinan Huanghe River Crossing tunnel construction.

At ring 685, after the machine stopped, an anti-clogging agent was used, and the cutting head was rotated to deal with the cutting head clogging. The shield tunneling was renewed 24 hours later, and clear changes of the driving parameters occurred. The cutting head torque and the total thrust declined and then remained stable. As mentioned above, the cutting head torque, together with the total thrust of the machine, is closely related to the penetration depth per revolution. In the same way, two composite parameters of total thrust/penetration depth and cutting head torque/penetration depth are employed to re-plot the data in Figures 10 and 11. The results are given in Figure 12. From the development of the two new parameters, the clogging at rings 675–685 and the effects because of the countermeasures taken at ring 685 can be more clearly identified. The newly-defined parameters help confirm the existence of the cutting head clogging throughout driving the shield machine and can be used to evaluate the occurrence of the cutting head clogging.

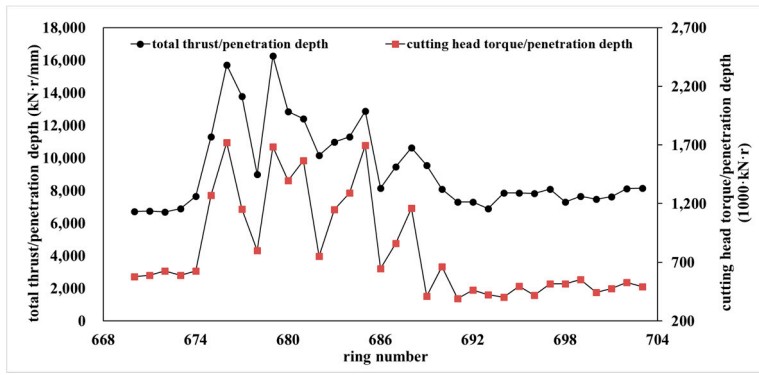

**Figure 12.** Changes of the two composite parameters at the Jinan Huanghe River Crossing tunnel construction.

## 4. Cutter Head Clogging at the Wuhan Metro Line 8 Yangtze River Crossing Tunnel Construction

The Wuhan Metro Line 8 Yangtze River Crossing tunnel was constructed using a Herrenknecht Mixshield with an excavation diameter of 12.55 m. To facilitate fast and safe cutting tool changes even under high pressure, the shield machine's cutting wheel was designed to be accessible in atmospheric conditions, resulting in an opening area ratio of about 28.5% of the cutting head, as shown in Figure 13.

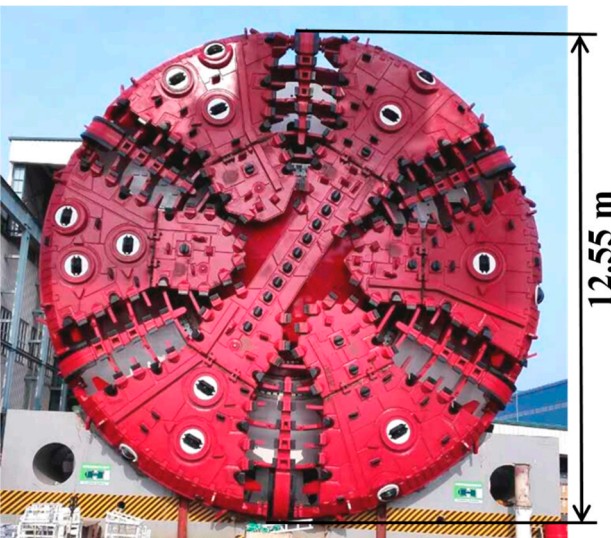

**Figure 13.** The cutting head used at the Wuhan Metro Line 8 Yangtze River Crossing tunnel construction.

As shown in Figure 14, the soils encountered within the tunnel alignment should be mainly fine sand, according to the geological report. But large amounts of clay grains and red clay were found in the excavated soils, as shown in Figure 15. The existence of the clay grains and the red clay undoubtedly increased the likelihood of clogging. As shown in Figure 16, the advance rate at rings 276 and 278 experienced great reductions, notwithstanding that fluctuations of the total thrust and the cutting head torque of the machine were within the accepted levels, as presented in Figure 16. It was guessed that cutting head clogging might occur. At ring 280, the machine ceased working, and chamber entrance work was performed to validate the assumption. The cleaning required for the cutting head clogging lasted 20 days. Rings 282 and 284 showed an improvement in machine performance, and a large increase in the advance rate of the machine.

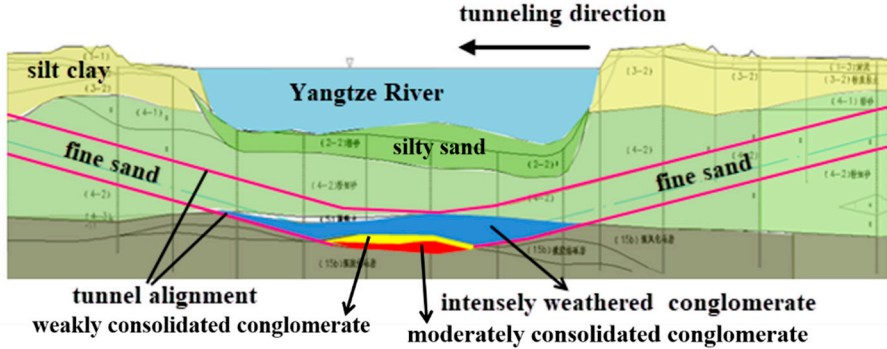

**Figure 14.** Longitudinal cross-section of the Wuhan Metro Line 8 Yangtze River Crossing tunnel.

To better understand changes of the main driving parameter, the two composite parameters of total thrust/penetration and cutting head torque/penetration were employed to re-plot the curves in Figures 16 and 17, resulting in the new curves shown in Figure 18. In observing the new curve changes, the occurrence of clogging at ring 279 and the effects because of the cleaning work at ring 280 can be more easily and intuitively identified. The two composite parameters help expose the existence of cutting head clogging, and can be used to evaluate the clogging.

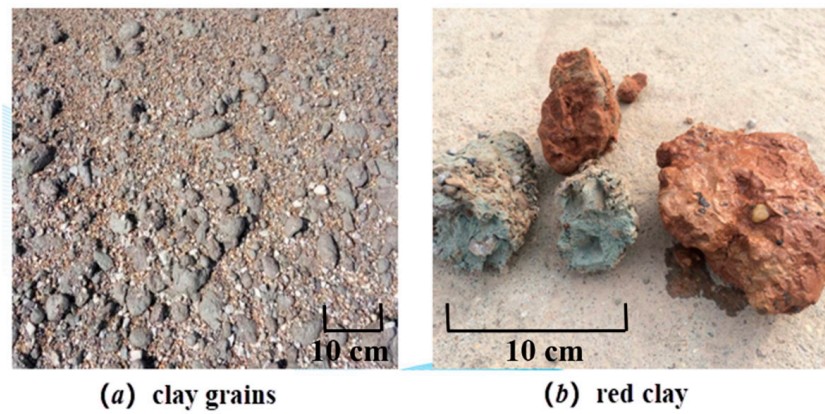

**Figure 15.** Clay grains and red clay in the excavated soils.

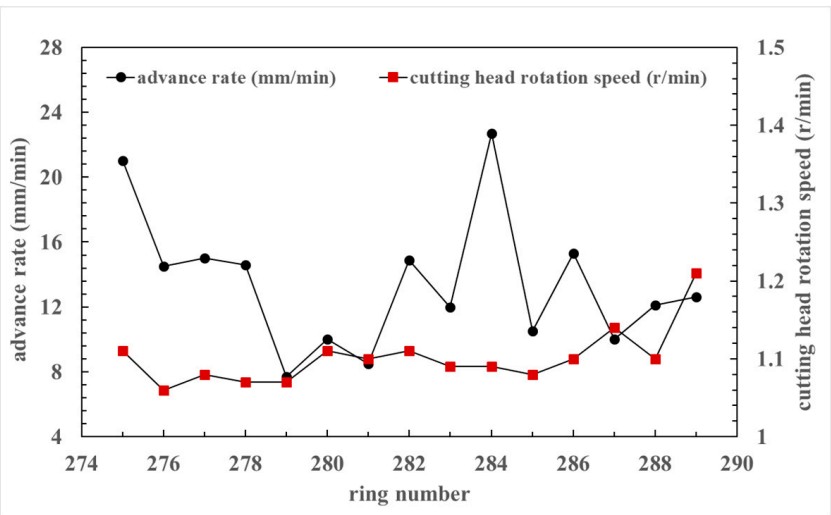

**Figure 16.** Changes of advance rate and cutting head rotation speed of the machine at the Wuhan Metro Line 8 Yangtze River Crossing tunnel construction.

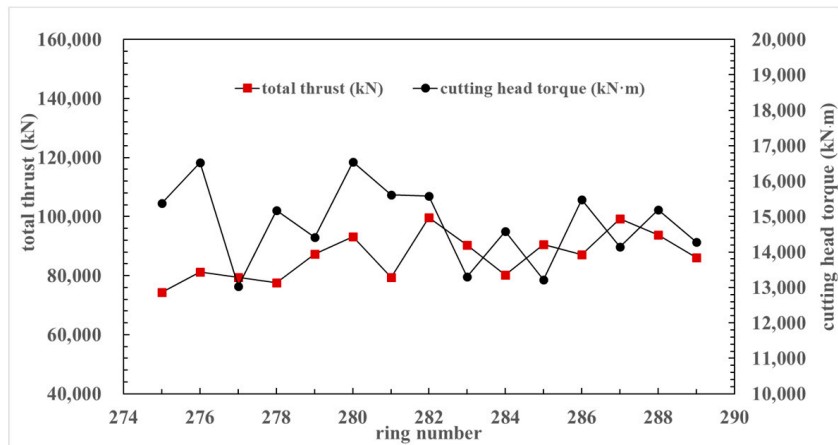

**Figure 17.** Changes of total thrust and cutting head torque of the machine at the Wuhan Metro Line 8 Yangtze River Crossing tunnel construction.

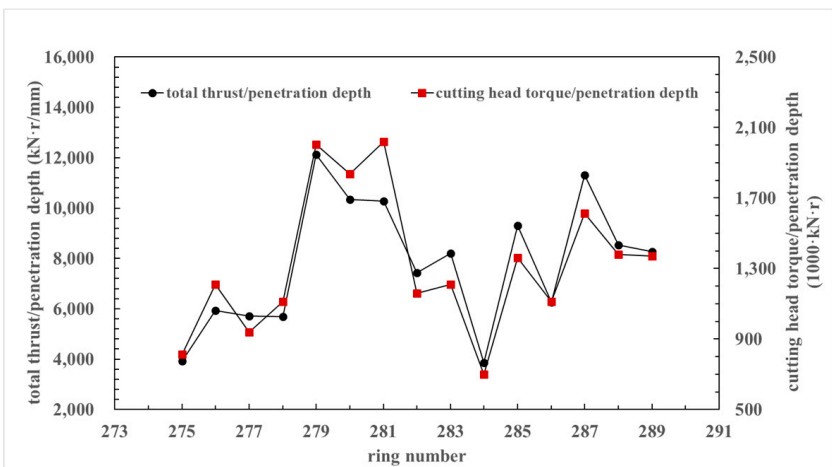

**Figure 18.** Changes of the two composite parameters at the Wuhan Metro Line 8 Yangtze River Crossing tunnel construction.

## 5. Discussions

(1)　Parameters influencing total thrust and cutting head torque of a machine

The necessary total thrust and cutting head torque of a shield machine is bound up with its dimension, shape, structure, as well as the tunnel burial depth and the soils (soft, hard, or both) within the tunnel alignment. For a tunneling machine, its advance rate and cutting head rotation speed also contribute to the actual total thrust and cutting head torque on the machine. A higher advance rate and slower cutting head rotation speed mean a higher penetration depth that requires higher cutting head torque and bigger total thrust [16]. For a tunneling machine, the soils that will be excavated and the tunnel alignment are predetermined, and the only choice to change the total thrust and the cutting head torque is to alter the penetration depth per revolution by changing the advance rate and the cutting head rotation speed. Moreover, when analyzing variations of the total thrust and the cutting head torque within a small tunneling range, considering the ground conditions, i.e., the soils and the tunnel burial depth to be constant in uttermost cases is reasonable. Therefore, when referring to alternations of the total thrust and the cutting head torque of a tunneling machine, the parameter of penetration depth per revolution does play a vital role and must be considered.

(2)　Performance of machines due to cutting head clogging

As mentioned above, the total thrust and the cutting head torque of a machine have something to do with the machine structure, particularly the cutting head structure. With the excavated soils flowing into the excavation chamber through the cutting head opening, the shield machine advances with driving parameters fluctuating within a normal range. For a machine with cutting head clogging, the opening area ratio of the cutting head declines, even to zero, causing difficulty in entrance into the excavation chamber of the excavated soils and causing the rapid increase in the total thrust and the cutting head torque of the machine. The scenarios were found at the Beijing south-to-north water diversion auxiliary project and presented in Figure 4.

When the cutting head opening is partially but not totally blocked, the excavated soil can still enter into the excavation chamber but at a slower speed. In this case, the total thrust and the cutting head torque might not experience rapid increases but retain normal fluctuations. The most influential is the decrease in penetration depth per revolution, which is the quotient of the advance rate and the cutting head rotation speed. This scenario happened at the Wuhan Metro Line 8 Yangtze River Crossing tunnel construction and is presented in Figures 16 and 17. That is to say that the total thrust and the cutting head torque of a tunneling machine perhaps do not rise rapidly and remain basically unchanged if its cutting head opening is only partially blocked. The results are heavily dependent

on the resulting penetration depth per revolution. The parameter of penetration depth considered, two composite parameters (total thrust/penetration depth and cutting head torque/penetration depth) are used to characterize the load development on the tunneling machines. The re-plotted curves are listed in Figures 7, 12 and 18. From these curves, the occurrence of cutting head clogging can be more easily identified, and the effect on account of the countermeasures taken to cope with the cutting head clogging is also more clearly displayed. Most importantly, the curves concerning the total thrust and those of the cutting head torque are more alike. A single curve is sufficient to determine whether cutting head clogging occurs or not.

(3)    Increased ranges of the two composite parameters of the clogged machine

Due to soils sticking to the cutting head surface and blocking the cutting head opening, the total thrust of the machine and the cutting head torque will increase if its advance rate does not decrease (usually, the cutting head rotation speed remains constant). The increase will continue, and a sharp increase will appear if the cutting head opening is completely blocked, meaning no excavated soils enter the excavation chamber.

The increased ranges of the two composite parameters are most closely related to the cutting head structure, especially the open area ratio of the cutting head. Of course, the penetration depth per revolution of the cutting head also contributes to the increase. The larger the cutting head opening area ratio, the larger the increased range of the redefined thrust or the redefined cutting head torque. This is true of the penetration depth per revolution of the cutting head. From the results of the above three construction cases presented in Figures 7, 12 and 18, the maximum increase by 2–6 times can be identified. As far as the minimum increase of the thrust/penetration depth and the torque/penetration depth of a clogged machine is concerned, this task, though challenging, is of vital importance. Given that most cutting head opening area ratios lie within the range of 30–50%, a 30–50% increase of the composite parameters should be paid attention to, especially when shield tunneling in soils liable to clogging. Also, chamber entrance, if allowed, to evaluate the condition of the cutting head is necessary to decide whether more countermeasures are taken.

## 6. Conclusions

The traditional methodology of evaluating clogging potential relying only on soil types is sometimes unrealistic and impractical due to the soils and properties of the soils being unknown. In this study, criteria for cutting head clogging during soft ground shield tunneling were established based on the machine's main driving parameters. Three slurry shield tunneling cases concerning cutting head clogging are introduced for validation. Changes in driving parameters of the machines, including total thrust, cutting head torque, advance rate, and cutting head rotation speed—before, during, and after the cutting head clogging—are presented and analyzed. Based on the analysis, the criteria for cutting head clogging occurrence during slurry shield tunneling are suggested. The conclusions that can be drawn are as follows:

(1)    When evaluating the total thrust and the cutting head torque of a shield tunneling machine, the parameter of penetration depth per revolution should be considered.

(2)    The total thrust and the cutting head torque of tunneling machines will increase, or will not, once cutting head clogging occurs. Either of the composite parameters of total thrust/penetration depth and cutting head torque/penetration depth is suggested to do the cutting head clogging evaluation.

(3)    According to the three construction cases presented, maximum increases of the total thrust/penetration depth and cutting head torque/penetration depth (suggested criteria) by 2–6 times can be found when cutting head clogging occurs. But regarding the minimum, a 30–50% increase of the composite parameters requires attention.

**Author Contributions:** X.L. (Xinggao Li) and Y.Y.: conceptualization, methodology, project administration, and writing the original draft. X.L. (Xingchun Li) and H.L.: data collection, plotting curves and analysis, and review. All authors have read and agreed to the published version of the manuscript.

**Funding:** Support from the Chinese National Natural Science Foundation (No. 51978040) is acknowledged.

**Institutional Review Board Statement:** Not applicable.

**Informed Consent Statement:** Not applicable.

**Data Availability Statement:** The data used to support the findings of this study are available from the corresponding author upon request.

**Conflicts of Interest:** The authors declare no conflict of interest.

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
