# Peer review of "Criteria for Cutting Head Clogging Occurrence during Slurry Shield Tunneling"

_applsci, doi:10.3390/app12031001_

Round 1

Reviewer 1 Report

The paper “Criteria for cutting head clogging occurrence during slurry shield tunnelling” reports an interesting work about the evaluation of the criteria for cutting head clogging occurrence based on the machine driving parameters. As highlighted in the text, the topic of the manuscript is characterized by a great engineering interest, especially for practical applications. The proposed method is validated through the application to three different case studies, all of great interest: (i) the Beijing south to north water diversion auxiliary project, (ii) the Jinan Huanghe River Crossing tunnel construction and (iii) the Wuhan Metro Line 8 Yangtze River Crossing tunnel construction. In fact, it is opinion of this reviewer that the paper can be published in Applied Science Journal after the following minor corrections/improvements:

  • rewrite the paper according to the Applied Science template and the journal guidelines
  • do not use abbreviations in the text as wasn’t, doesn’t…
  • In figure 6 a parenthesis is missing (kN r/mm
  • uniform the size of the graphs
  • improve the Section 6 (Conclusion) highlighting the innovative aspects of the work

Author Response

The authors thank the editor and reviewers for the time and attention in handling our manuscript.

We sincerely thank the reviewers and reviewers for taking the time and effort to review our manuscript, raise key issues and questions, and provide valuable comments and feedback. Guided by the reviewers’ questions and feedback, the manuscript has been significantly revised. Our responses to the reviewers’ specific comments are presented below.

Note: All key changes are highlighted in red color in the revised manuscript. All lines mentioned are referring to the revised manuscript.

Reviewer 2 Report

Dear Authors,

Thank you for the very nice and the well illustrated document to a very important topic in practice!

Only some few notes to the document:- Abstract: line 9: better write "It is recommended to take two combined parameters into account..", combined instead of composite
- shortly describe technical terms (e.g. plastiity index, consistency index,..) when these are used for the first time.
- References: are solely found in the introduction, the main text should also be substanitated by some references; are there some newer references available to this topic?
- End of introduction: mention the three projects, maybe add a simple geological overview map where the location of the three tunnels is shown
- chapter 2, second paragraph: use 12.8 m instead of 12.84 m (feigns a non-existent accuracy), same in line below Fig. 7
- Fig. 1: kilometre marking is not readable, legend is missing
- for all figures: reference at the illustrations is missing!
- Fig. 6: bracket is missing
- Fig. 13: legend missing (not all rock types are labelled)
- Fig. 15: use same formatting as in figures before (e.g. Fig. 9-11)
- Conclusions: first line: "...are introduced.."; under (3) mention the suggested criteria shortly again

Best regards

Author Response

(The authors gave the same response as above.)

Reviewer 3 Report

  1. The article presents interesting cases of tunneling and problems with clogging of the shield of a tunnel machine.
  2. It is advisable to use colors sequentially for the two variables presented in figures 15, 16 and 17 (similar to the previous figures).
  3. It would be good to present the composition  and parameters of an agent eliminating the problem of clogging of the disc, if it is not the subject of a patent.

Author Response

(The authors gave the same response as above.)
